# Visuoinertial and visual feedback in online steering control

Jo-Yu Liu[1]* , James R. H. Cooke[1] , Luc J. P. Selen[1], W. Pieter Medendorp[1]

Radboud University, Donders Institute for Brain, Cognition and Behaviour, Nijmegen, Netherlands

☺ These authors contributed equally to this work.

* joyu.liu@donders.ru.nl

## Abstract

Multisensory integration has primarily been studied in static environments, where optimal integration relies on the precision of the respective sensory modalities. However, in numerous situations, sensory information is dynamic and changes over time, due to changes in our bodily state and the surrounding environment. Given that different sensory modalities have different delays, this suggests that optimal integration may not solely depend on sensory precision but may also be affected by the delays associated with each sensory system. To investigate this hypothesis, participants (n = 22, 16 female) engaged in a continuous steering task. Participants sat on a motion platform facing a screen that displayed a cartoonish traffic scene, featuring a car traveling along a road. In the visuoinertial condition, where vestibular and somatosensory feedback were available, they were tasked with counteracting an external multi-frequency perturbation signal, which laterally perturbed the platform and the car, such that the car was kept within the center of the road. In the visual condition, the visual car was perturbed, while the motion platform remained stationary. We show that participants compensate better for the perturbation in the visuoinertial than the visual condition, particularly in the high frequency range of the perturbation. Using computational modelling, we demonstrate that this enhanced performance is partially due to the shorter delay of the vestibular modality. In this condition, participants rely more on the vestibular information, which is less delayed than the more precise but longer delayed, visual information.

**Data availability statement:** All data, modelling and statistics scripts are provided with open access at https://data.ru.nl/login/reviewer-3190997070/O65BDJR3IN77LQXS25HIR62YOE7FNAF5VI2XJQA.

## Author summary

Navigating the world effectively requires the continuous integration of sensory information related to our self-motion, forming a cohesive representation of our current location and direction. In real-life situations, the sensory environment changes continuously, as does the state of our body. In such scenarios, how are sensory inputs combined? We developed an experimental paradigm where participants were tasked with controlling a simple vehicle while counteracting a time-varying perturbation signal. To assess the

**Funding:** Our research is funded by the Dutch Research Agenda NWA (https://www.nwo.nl/en/researchprogrammes/dutch-research-agenda-nwa, NWA-ORC-1292.19.298), specifically, the NWA ORC ACT consortium (https://nwa-act.nl/), awarded to our principal investigator WPM. JYL and JRHC received salary support from an NWA grant (NWA-ORC-1292.19.298), awarded to WPM. WPM is additionally supported by NWO-SGW-406.21.GO.009 and Interreg NWE-RE:HOME. The funders had no role in study design, data collection and analysis, decision to publish, or preparation of the manuscript.

**Competing interests:** The authors have declared that no competing interests exist.

role of different sensory inputs we manipulated whether visual and inertial (primarily vestibular) cues were available or solely visual cues. Our results show that the presence of both visual and inertial cues significantly improved task performance. Modeling suggests this improvement is partially likely due to the shorter delay associated with the vestibular system compared to the visual system. Collectively, our results suggest that sensory precision alone is insufficient for effective state estimation and control in naturalistic scenarios, where accounting for the different delays of the sensory systems is critical for optimal task performance.

## Introduction

When navigating the environment, such as when we walk or steer a car, integration of multiple sensory cues plays a fundamental role. Our perception of self-motion relies mainly on visual cues—like optic flow—as well as inertial sensory information from the vestibular system, and possibly the somatosensory system as well. The vestibular system, comprising of the otolith organs and semicircular canals, utilizes mechanotransduction mechanisms to sense head movements. Although each sensory modality provides information about self-motion, the quality of their information differs in precision and time scale [1–4]. That is, the visual and vestibular systems operate in different frequency ranges: the first detects sustained movements, best described as a low-pass filter, whereas the latter senses accelerative movements, exhibiting high-pass dynamics.

In neuroscience laboratories, the perception of self-motion from sensory cues has typically been tested using open-loop psychometric tasks, such as two-alternative forced-choice tasks [3,5–7]. In such tasks, perception is probed only at the end of each trial without feedback or time constraints. Under these conditions, results show that humans adapt their weighting of visual and vestibular information from trial to trial in proportion to their reliability [7–9]. For example, when visual precision is low, the weighting of the vestibular signal increases in self-motion perception [3,7]. By exploiting differences in sensory precision, the resulting integrated perception is optimal in terms of minimizing overall variance. Although the visual and vestibular system each demonstrate better precision at specific frequencies, it has been shown that they still exhibit optimal integration across a broad range of frequencies [10].

This process of optimally adjusting sensory contributions to perception based on sensory reliability is a hallmark of Bayesian state estimation, the currently dominant framework used to explore multisensory integration [11–14]. However, while it elegantly explains the mechanism underlying optimality in perception, much of the investigations are conducted under highly constrained laboratory conditions, where the sensory cue statistics are typically stationary and perception is examined in the absence of action. As a result, the associated Bayesian perception models are also static in their output.

In contrast, naturalistic state estimation is a dynamic process that occurs in a parallel, continuous, and tightly coupled loop with action. Controlling a closed-loop behavior, like steering a car, does not only have to deal with changing sensory statistics, but also with the feedback delays and dynamics of the sensory systems. For example, Rosenberg et al. [15] asked participants to balance a joystick controlled chair, on which they were seated, and showed that a static measure of vestibular sensory precision correlated with dynamic task performances. Furthermore, in closed-loop control, the dynamics of the neuromuscular system play a crucial role when converting perception into action [16–19].

It has been argued that when the motion is self-generated, self-motion estimates not only depend on sensory and neuromuscular dynamics but can also be derived from internal models of closed-loop control dynamics developed from past experiences, which transform motor commands into predicted sensory consequences [11,20–23]. In support, Alefantis et al. [24] found that participants were able to actively navigate the environment with optic flow cues, but also without any sensory feedback, based on an internal model of the control dynamics. Recently we have shown that humans benefit from predictable changes in the sensory dynamics when controlling their motion by steering a motion platform, suggesting they can learn and rely on an internal model in self-motion estimation [25]. However, since these experimental tasks did not necessitate performance under time pressure, they were not appropriate for assessing the impact of sensory delay on the control loop.

In limb feedback control, Crevecoeur et al. [16] found that proprioceptive information is weighted more than visual information under conditions that demand rapid responses. In addition, Shayman et al. [4] showed that the ability of temporally combining visual and vestibular cues is influenced by their respective noise levels, but this study focused on perception, not closed-loop control. The goal of the present study is to examine the role of inertial and visual contributions to active steering control.

Here, building on previous work from Nash and Cole [26], we formulated an optimal feedback control model, incorporating visual, vestibular, and neuromuscular dynamics, noise and delays, as well as an internal model, to derive the optimal control actions in a steering task. Guided by this model and its predictions, we designed an online steering task in which participants must actively null out pseudorandom motion perturbations (0.1-1.93 Hz) to make a car visually stay on the road in front of them.

## Materials and methods

### Ethics statement

The study was approved by the ethics committee of the Faculty of Social Sciences of Radboud University Nijmegen, the Netherlands (Approval number: ECSW-2022-082R1).

### Participants

Twenty-two naive participants (6 men, 16 women), ranging in age from 18 to 33 years, without history of motion sickness, gave written informed consent to participate in the study. They were compensated with course credits or €15.00. The experiment lasted approximately 90 minutes per participant.

### Setup

Participants were seated comfortably on a custom-built linear motion platform (dubbed the vestibular sled), where the motion axis is aligned with their interaural axis (see Fig 1A). Their head was restrained. They were further secured by a five-point seat belt in combination with vacuum cushions molded around the torso, as well as straps around their legs and feet. The track of the sled is 0.9 m long. The sled is powered by a linear motor (TB15N; Tecnotion, Almelo, The Netherlands) and controlled by a servo drive (Kollmorgen S700; Danaher, Washington, DC). Emergency stop buttons on either side of the sled could be pressed to stop the experiment at any time. A steering wheel (Logitech G27 Racing Wheel, Lausanne, Switzerland) was mounted at a comfortable handling distance. The participants placed both hands on on the steering wheel at the 9 and 3 o'clock positions. The steering wheel, which had a rotational range of −450° to +450° and a resolution of 0.06°, controlled the speed of the

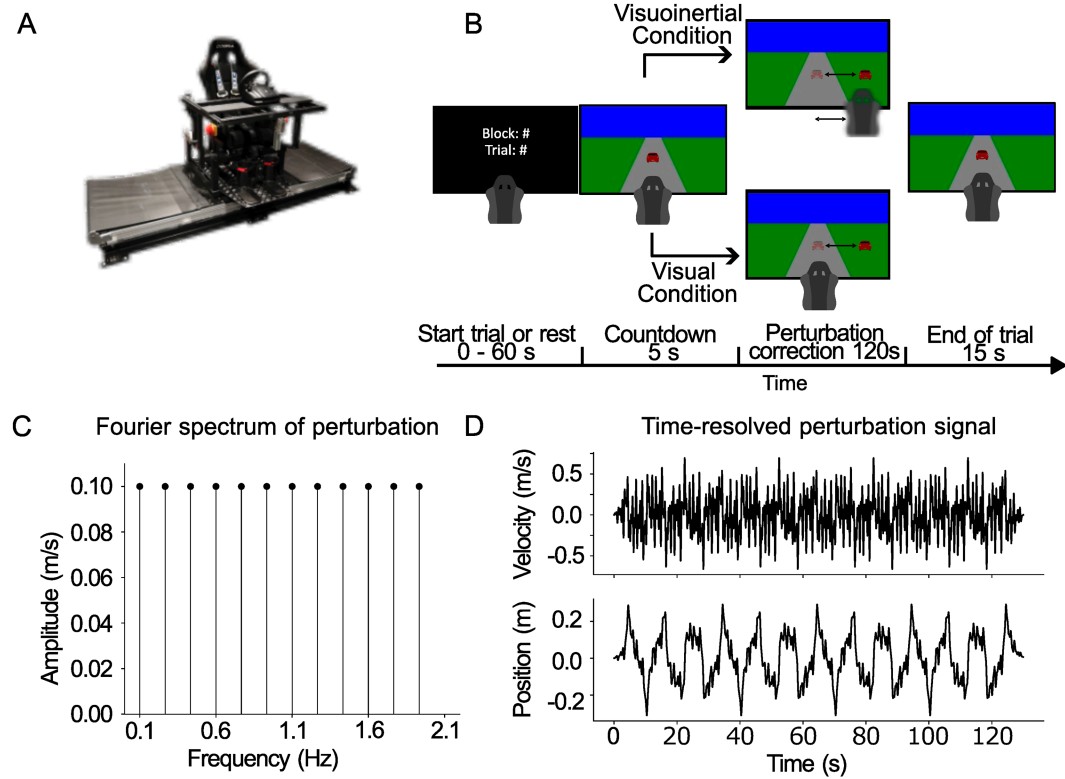

**Fig 1. Schematic of setup, task and perturbation signal. A**: Linear motion platform with the mounted steering wheel. **B**: Schematic representation of the steering task. Participants were asked to counteract a perturbation signal that either continuously displaced the platform and the visual car relative to the center of the road (visuoinertial condition) or solely displaced the car while the motion platform remained stationary (visual condition). **C**: Frequency spectrum of the sum-of-sines perturbation signal. The amplitude of all frequencies was set at 0.1 m/s and the phase of each sinusoid was randomly chosen between 0 and $2\pi$. **D**: Position and velocity time traces of a resulting perturbation signal.

sled at 60 Hz. Participants viewed a 55-inch OLED screen (55EA8809-ZC; LG, Seoul, South Korea) with a resolution of 1920x1080 pixels and a refresh rate of 60 Hz, positioned centrally in front of the sled at a distance of 1.6 m. The OLED screen depicted a cartoon traffic scenario, i.e., a car on a road (see Fig 1B). The road was generated as a trapezoid polygon with a central width of 0.15 m. The centrally presented car, visually representing the participant, had a width of 0.084 m. The experiment was controlled by custom code written in Python 3 [27] using the PsychoPy module (v.3.6.9 [28]).

## Behavioral task

Participants were seated on the motion platform in front of a screen, which depicted a cartoon traffic scenario, i.e., a car on a road (see Fig 1B). The car was perturbed laterally relative to the road and the task of the participant was to continuously adjust its position to keep it within the center of the road. Participants were tested in two conditions:

**Visuoinertial condition.** In this condition, the sled and visual representation of the car were perturbed synchronously, following a pseudorandom sum-of-sines motion profile. The task of the participant was to actively steer the sled to null out the motion perturbations to keep the car centered at the road as much as possible. Hence, from a sensory perspective,

participants were provided with both visual and inertial information to control their steering.

**Visual condition.** In this condition, the sled remained stationary while the visual representation of the car was perturbed relative to the road with a pseudorandom sum-of-sines motion profile. The task of the participant was to actively steer the wheel to compensate for this visual car motion and make the car stay centered at the road as much as possible. Hence, from a sensory perspective, participants are only provided with visual information to control their steering.

In both conditions, the perturbation was a pseudorandom velocity signal, constructed as a zero-mean sum of sines based on 12 independent sinusoids (0.1, 0.26, 0.43, 0.60, 0.76, 0.93, 1.10, 1.26, 1.43, 1.60, 1.76 and 1.93 Hz, Fig 1C), each with an amplitude of 0.1 m/s. Phases of the 12 sinusoids were randomly drawn between 0 and $2\pi$. The velocity of the total perturbation signal was limited to 0.6 m/s. If the signal did not fulfil this criterion, new phases for the sinusoids were drawn. Fig 1D gives an example of the perturbation signal in a single trial, showing the wide frequency content and the unpredictability of the perturbation.

The net seen and felt displacement depended on the participant's control through the steering wheel. The angle of the steering wheel was converted into a velocity signal (0.008 m/s/°) that was added to the external perturbing velocity signal.

Trials had a total duration of 145 s, with a 5 s countdown before the perturbation starts, and a 5 s ramp-up and 15 s ramp-down phase of the velocity perturbation signal at the beginning and end of the trial, respectively. The end of the 5 s countdown was followed by a short tone; the ramp-down phase was also accompanied by a countdown. The part of interest of the trial was the 120 s interval between the ramp-up and ramp-down phases, which was sufficient to quantify performance but short enough for the participant to focus on the visuoinertial and visual control task. Participants were tested blockwise on these conditions. Each condition block was comprised of 1 practice trial and 10 experimental trials, each trial was separated by resting periods of 1 minute. The purpose of the practice trial was to familiarize the participants with the steering movements and the task. The two conditions were performed in sequential order, separated by a 5-minute break. Conditions were counterbalanced across participants. Participants did not receive any feedback about their performance apart from the instantaneous visual and vestibular feedback during the task.

## Analysis

Data were processed offline in Python 3 [27], using the NumPy [29] and SciPy [30] modules. We analyzed a total of 440 trials (22 participants x 2 conditions x 10 trials). Of each trial, the ramp-up and ramp-down phase were discarded such that only the central 120 s were analyzed.

As overall performance measures, we computed the root-mean-square of the car position relative to the center of the road of each trial, as well as for the first 60 s and last 60 s to examine within-trial changes.

To obtain insight in the frequency-dependent steering dynamics, we examined the compensatory velocity signal in terms of gain and relative phase as a function of frequency. First, we fast Fourier transformed the controlled vehicle velocity and perturbation signals separately to obtain the amplitude and its associated phase at each frequency. Gain is then defined as the ratio of the controlled over the perturbed amplitude. Relative phase is defined as the phase difference between participant generated velocity and the perturbation velocity at each frequency, which we obtained by finding the angle between the two signals. Ideal performance

is defined by unity gain and a $-\pi$ ($-180°$) phase difference, i.e., perfectly counteracting the perturbation signal resulting in no motion of the car on the screen.

## Statistics

We performed a three-way repeated measures ANOVAs on the root-mean-squared-error (RMSE) of the car position to check for significant differences between the two conditions, changes in performance across trials and within-trial improvements. To investigate the frequency-dependent control strategies, we performed two two-way repeated measures ANOVAs on the gain and relative phase. Factors were the frequency (the 12 frequencies from the perturbation signal) and the two conditions. Paired t-tests, Bonferroni corrected, were used for post-hoc tests if significant main effects were observed.

## Modelling

We represent the steering control task as a perturbation-rejection task. More specifically, the goal of the task is to steer the wheel to keep the vehicle centered on the road despite perturbations that are either perceived visuoinertially (visual, vestibular and somatosensory feedback) or visually (visual feedback only). While the task did not employ a first-person view of the road—making it perhaps less ecologically valid than actual steering—correction for perturbations relies on visual and inertial cues, which are inherently noisy and subject to time delays. The velocity of the vehicle can be regulated based on the position of the steering wheel, an interaction that depends on the participant's neuromuscular system.

Following work from Cole and Nash [18], we formulated a optimal feedback control model to simulate this task (Fig 2). In brief (see S1 Appendix for details), the model consists of three components: the plant, the state estimator and the controller. The plant describes the true dynamics of the system, in our case, both the vehicle dynamics (Fig 2A) and the participant's neuromuscular system interacting with the steering wheel (Fig 2D). The vehicle part of plant is perturbed by a band-limited white noise signal with discrete frequencies. The transfer function of the motion sled is near unity for frequencies up to 2.5 Hz. A noisy time-delayed output of the plant is observed by the sensory systems (Fig 2B)—the vestibular and visual systems. The state estimator optimally estimates the state of the plant based upon the received sensory measurements and predictions based on an internal model of the plant (Fig 2C). The optimal motor command for the neuromuscular control of the steering wheel is computed via a control policy based on inputs from the state estimator and keeping the visual error at zero with minimal effort being the goal (Fig 2C).

More formally, we model the task using a linear quadratic Gaussian system (LQG [31]) operating at 60 Hz, as such, one time step ($\Delta_t$) is approximately 17 ms. Accordingly, the true state of the system evolves over time according to a linear system subject to Gaussian noise,

$$\mathbf{x}_{t+1} = A\mathbf{x}_t + B\mathbf{u}_t + V\boldsymbol{\eta}_t + V_\epsilon \boldsymbol{\epsilon}_t \tag{1}$$

where $\mathbf{x}_t$ indicates the state of the system (both the vehicle and neuromuscular states) at time $t$, $A$ is the state transition matrix (the passive system dynamics), $B$ is the state transition matrix of active dynamics, $\mathbf{u}_t$ is the motor command of the participant, $\boldsymbol{\eta}_t \sim \mathcal{N}(0, I)$ is a multivariate standard normal representing various noise components (e.g motor noise), $V$ is a matrix which transforms these noise components into the states and $\boldsymbol{\epsilon}_t$ is the perturbation applied by the experimenter. Note that our motion platform is velocity controlled, and as a result, we modeled our system as a velocity-controlled first order system. As a result of this choice, we had to explicitly incorporate the acceleration perturbations into the model, using

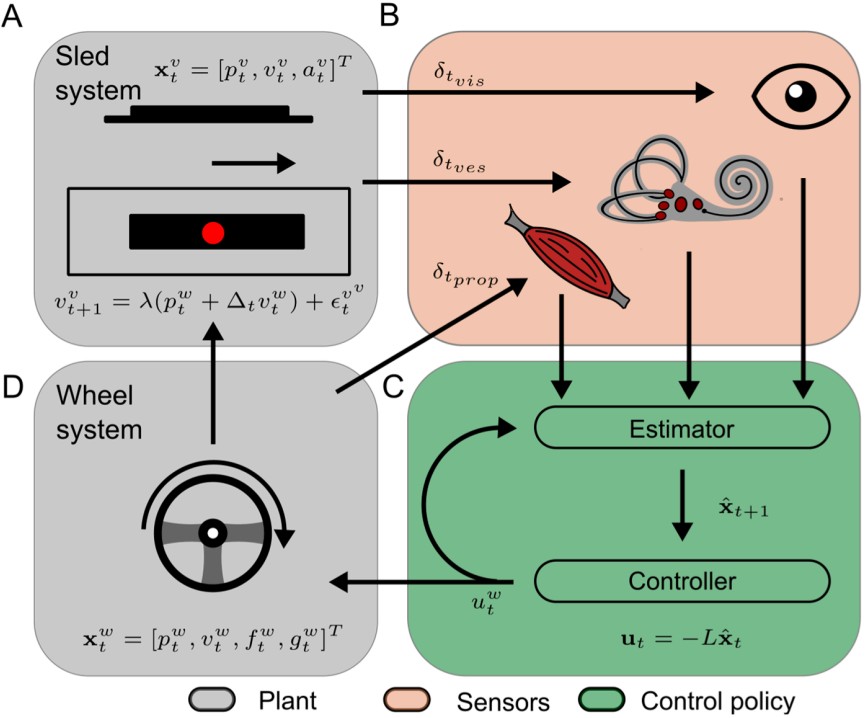

**Fig 2. Schematic model visualization.** In our model, the plant consists of the motion platform, screen (A) and wheel control system (D). Information is fed back through delayed and noisy visual, vestibular and proprioceptive sensory systems (B). Through the feedback, states of the system are then estimated with a Kalman filter (C). Finally, an LQR controller (C) generates an optimal control policy (i.e., motor command) to steer the wheel and motion platform.

a separate matrix $V_\epsilon$. This approach enables the model to detect changes not only in velocity and position, but also in acceleration.

For simplicity, we model the vehicle as a point mass with position $p_t^v$ and velocity $v_t^v$. Similar to the behavioural task we make the velocity of the vehicle a linear function (scaled by $\lambda$) of the wheel position $p_t^w$ and perturb it via $\epsilon_t^{v^v}$. Thus, in discrete time our vehicle dynamics are,

$$p_{t+1}^v = p_t^v + \Delta_t v_t^v \tag{2}$$

$$v_{t+1}^v = \lambda(p_t^w + \Delta_t v_t^w) + \epsilon_t^{v^v} \tag{3}$$

Because the wheel turns over a small range (45°) we also modelled the wheel dynamics as linear. This allows us to write the dynamics of the wheel as,

$$p_{t+1}^w = p_t^w + \Delta_t v_t^w \tag{4}$$

$$v_{t+1}^w = v_t^w + \frac{\Delta_t}{m} f_t^w \tag{5}$$

where the force ($f_t^w$) represents the angular force applied to the wheel through the arms. Thus, the force exerted on the wheel ($f_t^w$) is equivalent to the acceleration component of the control signal. To emulate the properties of the neuromuscular system [32], a second-order

low-pass filter is applied to the control prior to the conversion to force. This yields discrete time dynamics (note we split the second order filter to two first-order filters),

$$f_{t+1}^w = f_t^w \left(1 - \frac{\Delta_t}{\tau_{nm}}\right) + \frac{\Delta_t}{\tau_{nm}} g_t^w \tag{6}$$

$$g_{t+1}^w = g_t^w \left(1 - \frac{\Delta_t}{\tau_{nm}}\right) + \frac{\Delta_t}{\tau_{nm}} (u_t^w + \sigma_m \eta_t^{g^w}) \tag{7}$$

where $\tau_{nm}$ the time constant of the neuromuscular low-pass filter was taken to be 40 ms, similar to previous biological control models [32,33]. Finally, the otolith system perceives the physical vehicle acceleration so we also keep track of this in the state description. For simplicity, we compute this via finite difference (see S1 Appendix for the exact form),

$$a_{t+1}^v = \frac{1}{\Delta_t} (v_{t+1}^v - v_t^v) \tag{8}$$

Thus, the full state consists of the position, velocity and acceleration of the vehicle and wheel, alongside the neuromuscular states that drive the wheel, $x_t = [p_t^v, v_t^v, a_t^v, p_t^w, v_t^w, f_t^w, g_t^w]^T$. Rather than observing the true states, we assume the participant only has access to noisy and delayed observations of the states via the different sensory systems (e.g visual, vestibular and proprioceptive). The time delay of the visual and vestibular feedback are assumed to be approximately 120 and 10 ms, respectively, in the default model [34–36].

To this end, we assume the visual system provides measurements of the vehicle position and velocity delayed by 7 time steps ($\delta t_{vis} \approx 117$ ms),

$$y_t^{p^v} = p_{t-\delta t_{vis}}^v + \sigma^{p^v} \omega_t^{p^{vis}} \tag{9}$$

$$y_t^{v^v} = v_{t-\delta t_{vis}}^v + \sigma^{v^v} \omega_t^{v^{vis}} \tag{10}$$

So, the participant effectively visually observes a noisy version of position and velocity from a previous time point ($t - \delta t_{vis}$). Secondly, we assume that the vestibular and somatosensory system provides acceleration information during the visuoinertial condition (in the visual condition this observation is not available) but delayed by 1 step ($\delta t_{ves} \approx 17$ ms) that differs from $\delta t_{vis}$,

$$y_t^{a^v} = a_{t-\delta t_{ves}}^v + \sigma^{a^v} \omega_t^{a^v} \tag{11}$$

Finally, since the proprioceptive system of the arms do not provide direct information about the lateral translation of the car, we assumed that the position, velocity and force related to steering wheel were available with no delay ($\delta t_{prop} = 0$ ms),

$$y_t^{p^w} = p_{t-\delta t_{prop}}^w + \sigma^{p^w} \omega_t^{p^w} \tag{12}$$

$$y_t^{v^w} = v_{t-\delta t_{prop}}^w + \sigma^{v^w} \omega_t^{v^w} \tag{13}$$

$$y_t^{f^w} = f_{t-\delta t_{prop}}^w + \sigma^{f^w} \omega_t^{f^w} \tag{14}$$

This can be written more concisely in matrix notation as,

$$\mathbf{y}_t = H\mathbf{x}_t + G\boldsymbol{\omega}_t \tag{15}$$

where $\mathbf{y}_t = [y_t^{p^v}, y_t^{v^v}, y_t^{a^v}, y_t^{p^w}, y_t^{v^w}, y_t^{f^w}]^T$ is a vector of observations, $\boldsymbol{\omega}_t \sim \mathcal{N}(0, I)$ is a multivariate standard normal vector representing observational noise, $H$ is the observation matrix which maps from the true states to observations, and $G$ is a matrix which maps the noise to the observation. G involves a scaling constant $\sigma_s$ to scale noise levels without changing their relative magnitudes (see S1 Appendix for details).

From the standpoint of the participant, their goal is to determine the optimal motor command at each point in time to accomplish the task. To do this, it is necessary to specify a particular cost they wish to minimize which is indicative of their goals for the task. A crucial component of the task is keeping the car on the center of the road while minimizing the amount of effort to do so. In the model this is realized via a quadratic cost function which penalizes deviations of the vehicle's position from zero (with a weight of 1) and penalizes the magnitude of the control command (with a weight $r$),

$$J = \frac{1}{2}\mathbb{E}\left[\sum_{t=0}^{\infty}(x_t^p)^2 + r(u_t)^2\right] \tag{16}$$

This can be rewritten in terms of Eq (1) as,

$$J = \frac{1}{2}\mathbb{E}\left[\sum_{t=0}^{\infty}(\mathbf{x}_t^T Q \mathbf{x}_t + \mathbf{u}_t^T R \mathbf{u}_t)\right] \tag{17}$$

where $Q$ is a matrix reflecting the state costs which, in our task, is the positional deviation of the car from the road. $R$ is a matrix reflecting the cost of control (see S1 Appendix for their construction).

An ideal observer uses an internal model of the system (effectively, a model of Eqs 1 and 15), alongside the defined cost (Eq 17), to compute an optimal policy. However, the true state ($\mathbf{x}_t$) in Eq (17) is not available to the participants, so they have to use an estimate of this state ($\hat{\mathbf{x}}_t$). Because Eqs (1) and (15) constitute a linear system and Eq (17) is quadratic, the optimal policy [31] consist of using a Kalman filter to estimate the state. We utilized the filtering version [37] appose to the predictive version sometimes used in other control models [33]. First, the observer creates a prediction about the future state which is corrupted by noise, this prediction is then updated with the received measurement,

$$\hat{\mathbf{x}}_{t+1|t} = A\hat{\mathbf{x}}_t + B\mathbf{u}_t + \boldsymbol{\xi}_t \tag{18}$$

$$\hat{\mathbf{x}}_{t+1} = \hat{\mathbf{x}}_{t+1|t} + K(\mathbf{y}_{t+1} - H\hat{\mathbf{x}}_{t+1|t}) \tag{19}$$

and the control command is found by applying a linear feedback control rule to this estimate,

$$\mathbf{u}_t = -L\hat{\mathbf{x}}_t \tag{20}$$

where $K$ is the optimal Kalman gain and $L$ is the optimal feedback gain (see the S1 Appendix for details on how $L$ and $K$ are computed) and $\boldsymbol{\xi}$ is noise added to the prediction of the state estimator [37].

The overall behaviour of the model is dictated by a second-order transfer function for the neuromuscular dynamics, alongside the scaling of the various noise levels in the system (all modelled as Gaussian white noise), the relative control costs, and the sensory delays (as detailed in Fig 3 and derived in the S1 Appendix).

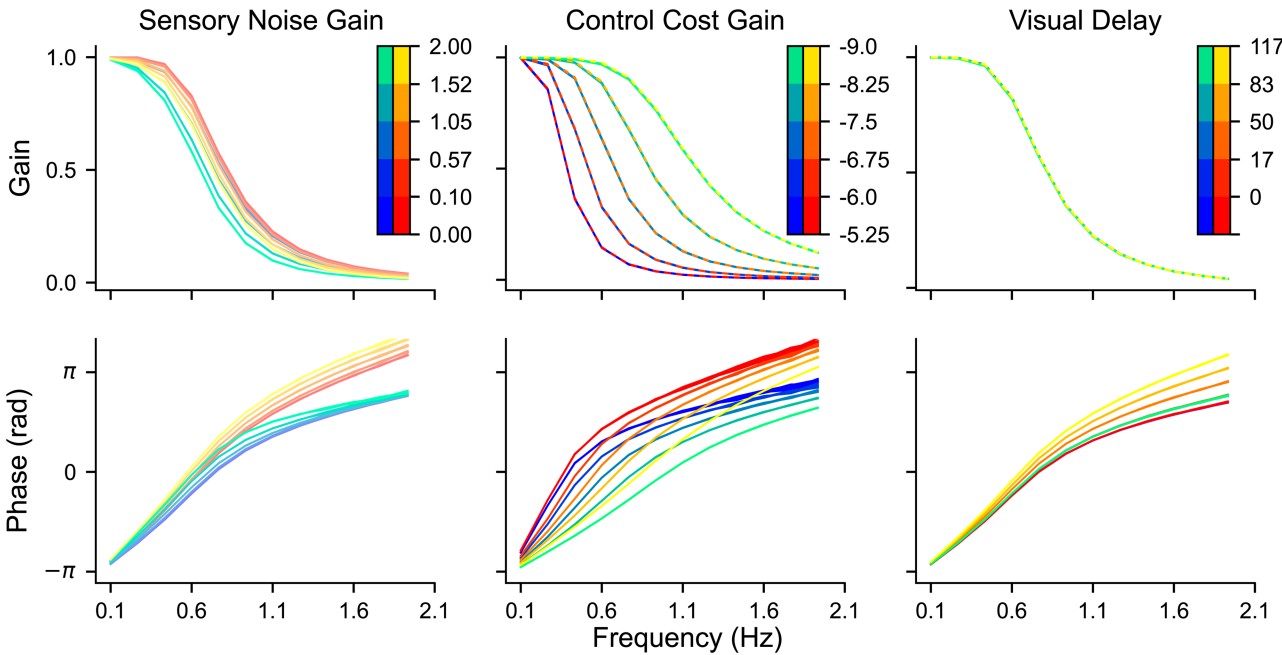

**Fig 3. Model predictions.** The upper row displays the model's gain predictions, while the lower row shows phase predictions. Each column (from left to right) illustrates the model's predictions as different parameters are varied: $\sigma_s$ (sensory noise gain, color-coded), $\mathbf{r}$ (control cost gain), and $\delta_{t_{vis}}$ (visual delay, in milliseconds). Blue-green lines represent the visuoinertial condition, and red-yellow lines represent the visual condition. Left column: Control cost gain is fixed at the average of the best-fit values across conditions (see Table A in S1 appendix), and visual delay is set to 7 timesteps ($\delta_{t_{vis}} \approx 117$ ms). Sensory noise gain is varied, as indicated by the color bar. Middle column: Sensory noise gain is held constant at its best-fit value, and visual delay is at 7 timesteps. Control cost gain is varied. Right column: Sensory noise gain is fixed at its best-fit value, and control cost gain is set to the average of the best-fit values. Visual delay is varied.

## Model fitting

Naturally the predictions of the model are dependent on the specific choice of parameter values used. We fixed all the parameters (see S1 Table) except for the overall sensory magnitude $\sigma_s$, and the control cost $r$. We allowed the control cost to differ between the visual and visuoinertial conditions. We parameterized control cost as $r = 10^{r^e}$, where $r^e$ is the fitted control exponent. Thus we have three free parameter $\theta = \{\sigma_s, r^e_{vis}, r^e_{visuo}\}$.

We optimized these parameters to minimize the RMSE between the observed steering wheel angle and the predicted steering wheel angle, similar to previous approaches in system identification [26].

Specifically, at each time point we compute the predicted steering wheel angle (given previous angles) from the model, and then RMSE between this and the observed wheel angle. In order to do so we follow previous inverse control approaches [38] and create a joint system of $\mathbf{x}_t$ and $\hat{\mathbf{x}}_t$ and utilize this to compute the expected prediction error. For the minimization itself we used Bayesian Active Direct Search (BADS) [39,40]. We ran the fitting procedure with three separate initializations and selected the parameter set with the lowest error.

## Model simulations

To understand the qualitative behaviour of the model, we performed simulations of the optimal feedback model and analyzed the central 120 s of the simulated time series in the same

way as the experimental data. We simulated the expected response of the system to the perturbations for a single subject (11 trials per condition). For each trial we computed the model spectra (gain and phase) from the simulated time series and averaged them. To determine the sensitivity of these spectra to the specific parameter values, we simulated the model for a variety of parameter values for both the visuoinertial and visual condition. Specifically, we varied the magnitude of the sensory noise ($\sigma_s$), control cost exponent ($r^e$) and the delay of the visual system ($\delta_{t_{vis}}$), as explicated in the caption of Fig 3.

Fig 3 illustrates the results of the simulations as bode diagrams for both the visuoinertial (blue-green lines) and visual condition (red-yellow lines). The general observation across both conditions is that of a low-pass system, i.e., with increasing perturbation frequency the gain reduces and the phase is increasingly less compensatory to the perturbation signal. There are three primary aspects of the model that could contribute to this low-pass property. The first is the LQR controller which operates as a low-pass filter when control costs are not negligible [41], the second is the estimator, which can also operate as a low-pass filter, and the third is related to the inherent low-pass neuromuscular dynamics. The effect of the estimator and controller can be seen directly from our model simulations in Fig 3. For the controller, lowering control costs reduces the low-pass nature of the system, moving it closer towards a pass-through system where the gain is 1, and the phase is closer to $-\pi$. To examine the role of the estimator, we manipulated both the sensory noise and visual delays. Lowering sensory noise increases the compensatory gain and lowers the phase shift, but its effect is limited by the low-pass nature of the plant and the controller. By contrast, when visual delays are reduced, the impact on compensatory gain is minimal, primarily influencing the phase shift in the visual condition rather than in the visuoinertial condition. In the visuoinertial condition, the delays introduced by the visual system are offset by the vestibular system. Thus, vestibular input helps decrease the phase lag in the response, allowing the estimate to more closely match the actual state and resulting in a response that is less out of phase. Importantly, none of these manipulations can address how much the dynamics of the neuromuscular system contribute to the low-pass nature of our model. To address this, we simulated a simpler version of the model without the low-pass neuromuscular dynamics (see S1 Appendix). This simpler model showed very little difference in compensatory gain, but showed a relative phase closer to $-\pi$, indicating that some of the observed phase difference comes from this property. Combined, this indicates that the primary factor influencing the frequency response characteristics of the model is the controller, while the neuromuscular system and the estimator play a lesser role.

## Results

We combined a steering task with computational modeling of the sensorimotor control loop. In the visuoinertial condition, in the presence of visual and inertial cues, the participant had to actively steer the sled to null out the motion perturbations to keep the car visually centered on the road. In the visual condition, in the presence of visual cues only, the participant's steering behavior only affected the visual position of the car, which also needed to be centered on the road.

### Task performance

Fig 4A shows the typical performance of a single participant in a selected part (30 s) of a visuoinertial and a visual trial, as well as the underlying perturbation signal. If the participant had not corrected for this perturbation, the car on the screen would move relative to the road

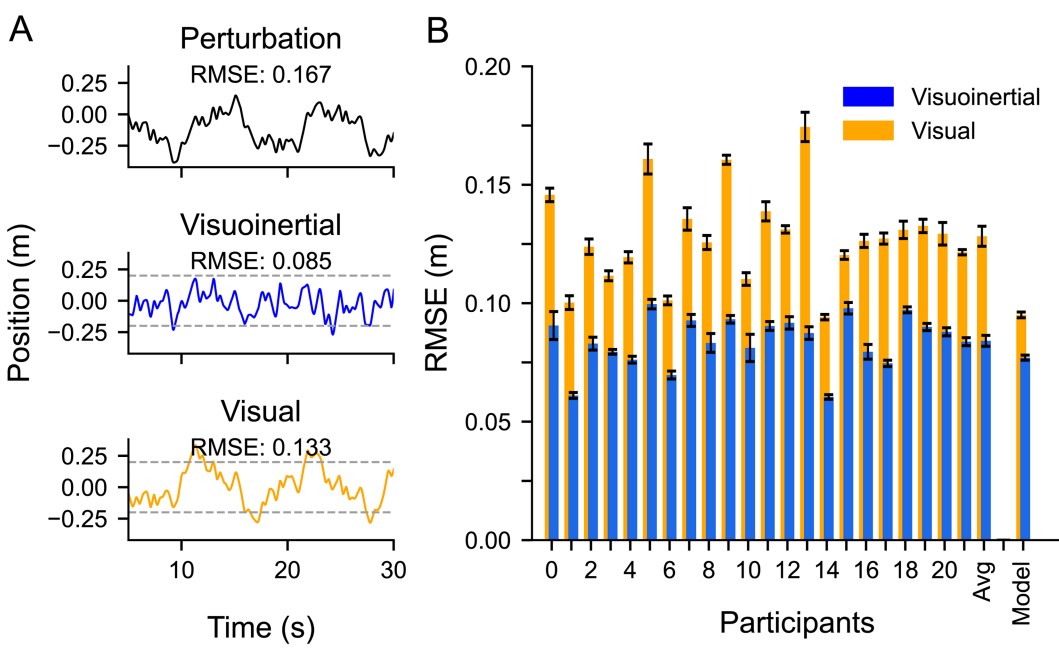

**Fig 4. Task performance, expressed as RMSE, for the visuoinertial and visual condition. A**: Part (30 s) of an example trial. Top: Uncorrected perturbation signal. Middle: Vehicle position in the visuoinertial condition. Bottom: Vehicle position in the visual condition. **B**: Average RMSE (±SE) separately for each participant and averaged across participants, as well as the average value predicted by the model (see Methods). Visuoinertial condition in blue, visual condition in orange.

along this profile, with displacements of more than 0.3 m relative to the center. In the visuoinertial condition (Fig 4A, middle panel), during which the participant sensed the motion of the car visually and vestibularly, there was a clear compensation for this perturbation. Compensation seems to be reduced in the visual condition (Fig 4A, bottom panel), where the participant does not receive vestibular cues, resulting in higher RMSE compared to the visuoinertial condition.

To quantify the overall performance in these trials, we calculated the zero-mean RMSE of the car relative the center of the road across the full 120 s period. While the RMSE of the perturbation was 0.167 m, this reduced to 0.085 m and 0.133 m in the visuoinertial and visual condition, respectively, due to active steering control by the participant.

These RMSE findings are exemplary for all participants. Fig 4B shows the RMSE for the two conditions across trials, separately for each participant, and averaged across participants. We ran a three-way repeated measures ANOVA with condition, trial and epoch (first and last 60 s of each trial) to examine their effects on the RMSE. First, we found a main effect of condition ($F(1,21) = 232.38$, $p< 0.001$, $\eta_p^2 = 0.92$), where the RMSE was on average 1.5 times smaller in the visuoinertial condition (RMSE = $0.08 \pm 0.01$ m (mean $\pm$ SD)) compared to the visual condition (RMSE = $0.13 \pm 0.02$ m). Interactions of this factor with trial or epoch were not significant (condition x trial: $F(9,189) = 1.63$, $p = 0.11$, $\eta_p^2 = 0.07$; condition x epoch: $F(1,21) = 3.47$, $p = 0.08$, $\eta_p^2 = 0.14$). Also the three-way interaction effect was not significant ($F(9,189) = 0.82$, $p = 0.59$, $\eta_p^2 = 0.04$).

The ANOVA model revealed no significant effect of trial number either ($F(9,189) = 0.93$, $p = 0.51$, $\eta_p^2 = 0.042$). Fig 5A provides an illustration of this result, suggesting there is no across trial learning in our paradigm. There was neither an interaction effect of trial with epoch (trials x epoch: $F(9,189) = 0.94$, $p = 0.49$, $\eta_p^2 = 0.04$).

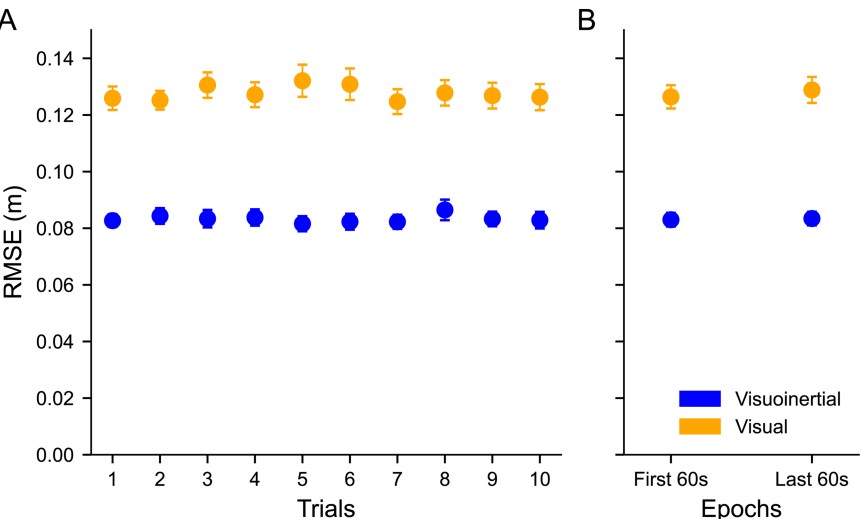

**Fig 5. RMSE changes within and across trials.** Error bars indicate the standard error of measurement. **A**: Across trial RMSE changes. No significant effect of trial was found, indicating no across trial performance changes. **B**: Within trial RMSE changes. The main 120 s portion of the trial is split into two 60 s epochs. Post-Hoc analysis revealed no significant differences between the first and second epoch in either condition, indicating no within trial performance changes.

Finally, the ANOVA yielded a significant main effect of epoch (F(1,21) = 5.56, $p$ = 0.03, $\eta_p^2$ = 0.21). This result is illustrated in Fig 5B. For the visuoinertial condition, both epochs had equal mean and standard deviation ($\mu_1,\mu_2$ = 0.083 m, $\sigma_1,\sigma_2$ = 0.011). The visual condition showed small differences ($\mu_1$=0.126 m, $\mu_2$=0.129 m, $\sigma_1$=0.018, $\sigma_2$=0.021). Given that the interaction effect between epoch and condition was close to statistical significance, we performed post-hoc paired t-tests for the visuoinertial and visual condition separately. This revealed neither a significant effect in the visuoinertial condition (t(21) = 0.45, $p$ = 0.66, Cohen's d = 0.03), nor significant effect of epoch in the visual condition (t(21) = -1.63, $p$ = 0.12, Cohen's d = -0.09).

These results suggest that participants have reached close to asymptotic performance levels after the practice trial, and that there were no significant performance improvements or deteriorations within a trial.

## Dynamics of compensatory control

To analyze the dynamics of the compensation that resulted in the RMSE differences between conditions, Fig 6A presents the gain and phase of the compensatory steering signal at the perturbed frequencies of the data (dots), averaged across participant, i.e., a Bode plot. Ideal performance would be represented by a gain of one and phase of $-\pi$ radians ($-180°$). The gain of the data (Fig 6A, top panel) clearly demonstrates lowpass behavior, mimicking the simulation observations seen in Fig 3. In the visuoinertial condition, the gain is about one for the lowest perturbation frequency, and reduces with increasing perturbation frequency. In the visual condition, the gain is above one for the lowest perturbation frequency, but rapidly drops below the level of compensation of the visuoinertial condition with increasing frequency. A two-way repeated measures ANOVA revealed significant main effects of both condition (F(1,21) = 32.69, $p$ < 0.01, $\eta_p^2$ = 0.609) and frequency (F(11,231) = 211.1, $p$ < 0.01, $\eta_p^2$ = 0.910), as well as a significant interaction between them (F(11,231) = 18.39,

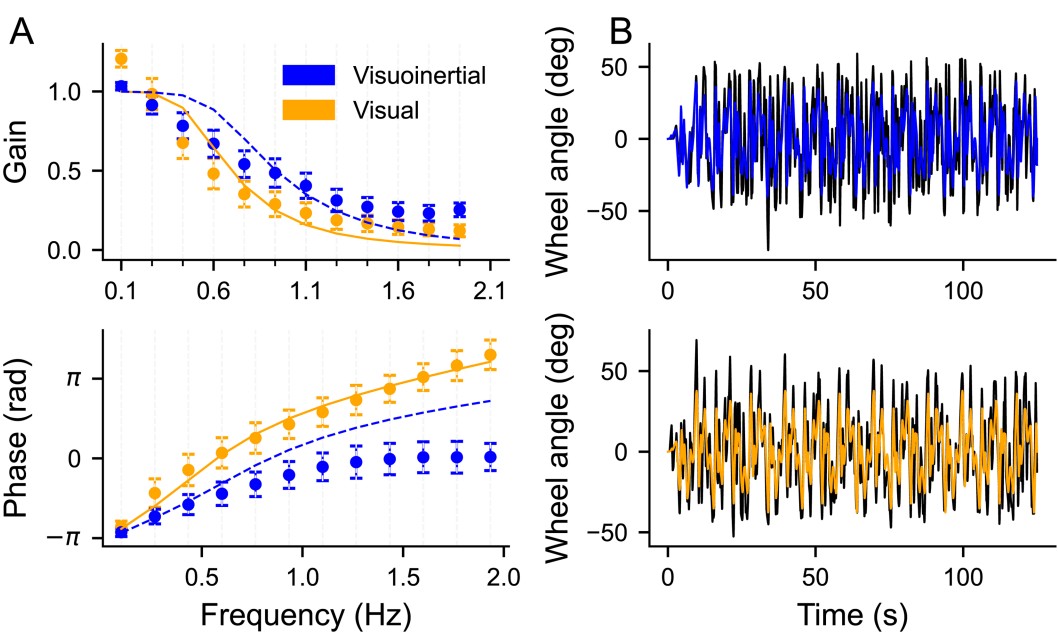

**Fig 6. Measured gain (top) and phase (bottom) alongside predictions. A**: Data alongside Model fits. Visuoinertial condition in blue; visual condition in orange. Data is plotted as as dots (mean ± SD), lines represent model predictions. **B**: Time series plots illustrating the steering response and model fit for a single trial from one participant.

$p < 0.01$, $\eta_p^2 = 0.467$). Post-hoc analysis revealed that the two conditions yielded different gains at all frequencies (t(21) > 2.41, $p < 0.03$, Cohen's d > 0.53) except at 0.26 Hz (t(21) = 1.46, $p = 0.16$, Cohen's d = -0.32). Note that, except for 0.1 Hz, all gains are lower in the visual condition.

For both conditions, the phase of the lowest frequency compensatory signal is about $-\pi$ radians, i.e., exactly in antiphase with, and thus compensatory for, the perturbation signal (Fig 6A, bottom panel). As the frequency increases, the phase increases as well, but much less for the visuoinertial than the visual condition. This is most likely due to the shorter sensory delay in the visuoinertial condition, i.e., the vestibular delay. A two-way repeated measures ANOVA showed a significant main effect of condition (F(1,21) = 1263.49, $p < 0.01$, $\eta_p^2 = 0.984$) and frequency (F(11,231) = 914.73, $p < 0.01$, $\eta_p^2 = 0.978$) as well as a significant interaction (F(11,231) = 428.4, $p < 0.01$, $\eta_p^2 = 0.953$). Post hoc paired t-tests revealed that the two conditions started to differ in phase for frequencies larger than 0.1 Hz (t(21) > 2.9, $p < 0.01$, Cohen's d < -2.42). At 0.1 Hz, no significant difference was found between conditions (t(21) = 0.75, $p = 0.23$, Cohen's d = -1.93).

To test how well our model can account for these data, we fitted its free parameters to the steering response in the time domain (see Methods). Fig 6B (top and bottom panels) illustrate that the fitted steering responses closely matched the measured responses for a representative participant in the visuo-inertial and visual-only condition, respectively. As shown in Fig 4B, the model fits closely match the observed RMSE in the visuoinertial condition but show a slight discrepancy in the visual-only condition. In term of fitted parameters, we found estimates of 0.5 cm visual position noise and 10 cm/s² of vestibular acceleration noise across participants, consistent with values reported in previous studies [42–44]. We also found that the fitted control cost exponent was negative and smaller (e.g control cost was higher) for the

visual condition compared to visuoinertial condition, suggesting participants were willing to apply more control in the visuoinertial condition.

Interestingly, when examining the frequency-resolved dynamics of the predicted compensation, the quality of the fitted pattern is slightly different. In the visual-only condition, the model provides a reasonably good fit for both the gain and phase data. However, when the same model is applied to the visuoinertial condition, it continues to fit the gain data well but tends to overestimate the phase compared to participants' actual performance.

## Discussion

We employed a neurocomputational framework in combination with a system identification approach to investigate the roles of vestibular and visual inputs in online steering control. Participants engaged in a perturbation-rejection task where they were required to steer a vehicle to maintain its position on the road despite perturbations that were either sensed visually (the visual condition) or in synchrony with inertial motion cues (the visuoinertial condition). Participants were better at stabilizing the vehicle in the visuoinertial compared to the visual condition (see Fig 4). Frequency analysis revealed that this better performance was associated with higher compensatory gains and lower phase shifts at higher frequencies in the visuoinertial than the visual condition (see Fig 6). To delve into the computational mechanisms, we developed an optimal control model that incorporated the dynamics of the involved sensory and motor systems and simulated the behaviour of this system in the two steering scenarios. The model provided a reasonable fit to the steering response data, although it performed better for the visual-only condition compared to the visuoinertial condition. The estimated free parameters were plausible, with visual position noise at 0.5 cm and vestibular acceleration noise at 10 cm/$s^2$. We will next connect our findings to earlier research and explore their significance for future studies.

Previous research in multisensory state estimation has shown that the integration of stationary sensory cues adheres closely to Bayes optimal rules across various sensory modalities [12–14]. This also includes the use of visual and vestibular information for heading estimation [3,7,35] as well as roll tilt estimation [10,11]. However, the paradigms employed do not capture the continuous, and often time-sensitive nature of state estimation. Rather, state estimation is conceptualized as a single two-alternative (2-AFC) response at the end of each trial, lacking time constraints and prioritizing accuracy over speed.

By including response time as a performance criterion, integration for state estimation appears to no longer obey the Bayesian integration rule that is employed in static experiments. For example, in heading perception, it has been shown that participants sacrifice precision to prioritize speeded responses [45].

Under continuous control, state estimation can be understood as the process of accumulating evidence over time. Given that each sensory system provides information about hidden states with different temporal characteristics, tasks that involve combined estimation and control of states should trade-off precise, but delayed, information for the earliest available information that is "good enough" [16,46]. This highlights a difference from conventional static multisensory research, where there has been a lack of investigations into how the inclusion of active components affects sensory integration strategies.

Recent studies addressed the gap between static and naturalistic self-motion state estimation by employing closed-loop path integration tasks [24,47,48]. Yet, these studies did not specifically evaluate the impact of sensory delay on path integration, as there were no limitations imposed on response time. Their findings revealed that the estimation of self-motion statistics during active path integration is derived from the continuous integration of optic

flow and vestibular feedback. Availability of optic flow patterns resulted in more accurate path integration than the presence of only vestibular cues, suggesting that visual cues are dominant in path integration. However, given that the vestibular system primarily responds to acceleration [49–51], the lower vestibular contribution could be due to the low and sustained speed by which participants navigated in these studies.

In the presence of velocity changes, i.e., accelerations, along with performance under time pressure, vestibular signals have been shown beneficial for steering control [52]. Nash and Cole [52] reported that vestibular cues significantly reduce driver's reaction times and path integration errors after an impulse perturbation during a driving task. Their single impulse perturbation paradigm allowed them to analyze their data in the time domain, quantifying performance in terms of compensatory response by aligning participant response with perturbation onset. In contrast, our use of a continuous, 120 s long, perturbation made it difficult to align participant responses with the perturbation within the time domain, but allowed us to use frequency-domain based analysis to quantify phase delay as the phase difference between participant response and perturbation. Nevertheless, their results are in line with ours, demonstrating the importance of considering delays in sensory systems in multisensory integration tasks where response speed is integral to performance. To our knowledge, theirs and our work is the first to show the effects of sensory delay on vestibular and visually-guided steering control under perturbations.

Time-critical performance is not only relevant in closed-loop steering but also for the ongoing control of standing balance [53–55]. When vision is the primary sensory channel of feedback in a quiet standing task (through limiting or removal of vestibular and somatosensory cues), postural movement becomes drastically more variable [56,57]. Artificially increasing the delays in the feedback loop further destabilizes balance [58]. Enhanced performance in vestibular conditions, whether related to balance or steering control, is likely attributed to differences in sensory delays between the visual and vestibular system. Nevertheless, caution is advised when drawing parallels between steering and balance control: balance is biomechanically unstable, steering is not. Furthermore, in the balance case, vestibular (and somatosensory) cues establish the reference system for the body vertical, although energy expenditure and control policy may also contribute to this set point. In contrast, in steering, only vision provides the reference system for road-centered position, while vestibular cues inform only about acceleration which is not a direct measurement of error.

We compared and interpreted our behavioral observations in relation to an optimal control model, based on a Linear-Quadratic-Gaussian (LQG) system. In this LQG model we made several assumptions about known and unknown hidden variables and control strategies of our participants. These assumptions may in part explain the discrepancies between our experimental observations and modeling outcomes. First, in the model, we assumed accurate knowledge, and thus veridical internal models of the dynamics of the sensorimotor system, the task, and motion platform. While our previous work suggests the brain can construct accurate internal model for steering [25], the neurophysiological evidence is still scarce [59]. Nonetheless, in the model, this knowledge of dynamics is assumed and represented by differential equations describing the dynamics of the motion platform and human sensorimotor system to make predictions about future states, and to generate the optimal control policy and state estimation gain. Additionally, we assume that the human brain has representations of the noise magnitudes and delays in the sensory and neuromuscular systems. In the model, we assume that noise operates in the system in an additive manner. While multiplicative noise has been suggested for motor processing [60], for computational tractability we assumed that the rotational range of the steering wheel is small enough to warrant additive noise sources.

For our model fit, we also assumed that all state representations are the same across the two conditions, aside from the observations made by the participant and the control cost ($Q$). However, this assumption may not hold, as there are substantial differences in the phase relationship between the two conditions that the model could not explain. The question remains: why do participants' response phases in the visual-inertial condition align more closely with the actual state than our model predicts? One possibility is that other parameters vary between the conditions that we have not factored in. Another explanation is that participants may employ estimation and control strategies different from those assumed in our model. We represented the perturbation as white noise, consistent with standard sum-of-sines system identification methods. However, participants could potentially detect the low-frequency periodic patterns within the perturbation and adapt their control strategies accordingly. For example, if participants are able to estimate the perturbation, the visuoinertial phase could be reduced, as vestibular input may be used to predict future states. Since there are numerous ways such estimation models could be constructed, a systematic investigation of these possibilities represents an intriguing direction for future research.

Finally, optimal feedback control has emerged as a prominent framework in sensorimotor control research, largely due to its capacity to demonstrate established principles such as Fitts' law and the minimum intervention principle [32]. For future work, one could raise the question whether these principles are equally applicable in steering control. For instance, to examine the applicability of the minimum intervention principle in steering control, one could adapt the via points reaching task from Liu and Todorov [61] into a steering-based parking scenario. Using our current paradigm, we could also further investigate the context dependent effects of control strategies that we speculated on. For example, we could manipulate the width of the road [62] and the corresponding goal to observe if participants adjust their control strategies accordingly.

In conclusion, our work highlights the essential function of the vestibular system in managing online steering control during unexpected perturbations. Specifically, participants demonstrated a greater ability to mitigate the impact of these disturbances when vestibular feedback was present. Our modelling suggests that this may be attributed to the shorter delay of the vestibular system compared to the visual system. There remains considerable work to be done to improve the alignment between the model predictions and the participant data. Lastly, we recommend conducting follow-up experiments to evaluate whether steering control adheres to the principles of optimal feedback control that have been shown in reaching behavior, and to test how context may effect control and estimation.

## Supporting information

**S1 Appendix. Detailed description of state space model and derivation of LQG control and estimator gain.** Discussion of frequency response properties of the model and its variants.
(PDF)

**S1 Fig. Frequency response of simplified models.** Frequency response of all models, simplified by making them fully observable and removing all noise aside from the perturbation. The purple line indicates the frequency response of a model without a neuromuscular low pass filter in the plant (thus controlling force). The blue line indicates the frequency response with the neuromuscular filter added in.
(EPS)

## Acknowledgments

JYL would like to thank Milou van Helvert and Sophie Willemsen for support with the motion platform. JYL expresses gratitude to the members of the SensorimotorLab for all their help and the stimulating discussions.

## Author contributions

**Conceptualization:** Jo-Yu Liu, Luc J. P. Selen, W. Pieter Medendorp.

**Data curation:** Jo-Yu Liu.

**Formal analysis:** Jo-Yu Liu, James R. H. Cooke, Luc J. P. Selen, W. Pieter Medendorp.

**Funding acquisition:** W. Pieter Medendorp.

**Investigation:** Jo-Yu Liu.

**Methodology:** Jo-Yu Liu, James R. H. Cooke, Luc J. P. Selen, W. Pieter Medendorp.

**Project administration:** Jo-Yu Liu, Luc J. P. Selen, W. Pieter Medendorp.

**Resources:** W. Pieter Medendorp.

**Software:** Jo-Yu Liu, James R. H. Cooke.

**Supervision:** James R. H. Cooke, Luc J. P. Selen, W. Pieter Medendorp.

**Validation:** Jo-Yu Liu, James R. H. Cooke, W. Pieter Medendorp.

**Visualization:** Jo-Yu Liu, James R. H. Cooke, W. Pieter Medendorp.

**Writing – original draft:** Jo-Yu Liu, James R. H. Cooke, W. Pieter Medendorp.

**Writing – review & editing:** Jo-Yu Liu, James R. H. Cooke, Luc J. P. Selen, W. Pieter Medendorp.

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
