## [Decision Letter · Decision Letter 0]

13 Feb 2025

PCOMPBIOL-D-24-02019

Visuoinertial and visual feedback in online steering control

PLOS Computational Biology

Dear Dr. Liu,

Thank you for submitting your manuscript to PLOS Computational Biology. After careful consideration, we feel that it has merit but does not fully meet PLOS Computational Biology's publication criteria as it currently stands. Therefore, we invite you to submit a revised version of the manuscript that addresses the points raised during the review process.

Please submit your revised manuscript within 60 days Apr 15 2025 11:59PM. If you will need more time than this to complete your revisions, please reply to this message or contact the journal office at ploscompbiol@plos.org. Please include the following items when submitting your revised manuscript:

We look forward to receiving your revised manuscript.

Kind regards,

Gunnar Blohm, Ph.D.

Academic Editor

PLOS Computational Biology

Daniele Marinazzo

Section Editor

PLOS Computational Biology

**Journal Requirements:**

https://journals.plos.org/ploscompbiol/s/figures  3) We have noticed that you have a list of Supporting Information legends in your manuscript. However, there are no corresponding files uploaded to the submission. Please upload them as separate files with the item type 'Supporting Information'.  4) Some material included in your submission may be copyrighted. According to PLOSu2019s copyright policy, authors who use figures or other material (e.g., graphics, clipart, maps) from another author or copyright holder must demonstrate or obtain permission to publish this material under the Creative Commons Attribution 4.0 International (CC BY 4.0) License used by PLOS journals. Please closely review the details of PLOSu2019s copyright requirements here: PLOS Licenses and Copyright. If you need to request permissions from a copyright holder, you may use PLOS's Copyright Content Permission form. Please respond directly to this email and provide any known details concerning your material's license terms and permissions required for reuse, even if you have not yet obtained copyright permissions or are unsure of your material's copyright compatibility. Once you have responded and addressed all other outstanding technical requirements, you may resubmit your manuscript within Editorial Manager.  Potential Copyright Issues:  - Figures 1A, 1B, 2B, and 2D. Please confirm whether you drew the images / clip-art within the figure panels by hand. If you did not draw the images, please provide (a) a link to the source of the images or icons and their license / terms of use; or (b) written permission from the copyright holder to publish the images or icons under our CC BY 4.0 license. Alternatively, you may replace the images with open source alternatives. See these open source resources you may use to replace images / clip-art: - https://commons.wikimedia.org - https://openclipart.org/. 5) Please ensure that the funders and grant numbers match between the Financial Disclosure field and the Funding Information tab in your submission form. Note that the funders must be provided in the same order in both places as well. State the initials, alongside each funding source, of each author to receive each grant. For example: "This work was supported by the National Institutes of Health (####### to AM; ###### to CJ) and the National Science Foundation (###### to AM).". If you did not receive any funding for this study, please simply state: u201cThe authors received no specific funding for this work.u201d 

**Reviewers' comments:**

Reviewer's Responses to Questions

**Comments to the Authors:**

Reviewer #1: In this manuscript, Liu and colleagues present an innovative closed-loop control task in which participants stabilize the position of a simulated car on a road. This task is implemented using a motorized sled under two conditions: visual-only and visuo-vestibular. The authors find that participants’ ability to control their trajectories under perturbations exhibits a low-pass dynamic, more pronounced in the visual-only condition than in the visuo-vestibular condition. This result aligns with prior knowledge that the vestibular system is more accurate than vision at high frequencies.

The authors also propose a detailed control model and attempt to fit it to the data. However, I have significant reservations about this model. Their attempt to fit both conditions with a single set of parameters (“Model Fit 1”) is largely unsuccessful, as this approach fails to reproduce the gain differences and only marginally explains the phase differences between the two conditions (Fig. 6A). They achieve better results by allowing the model to use two separate sets of parameters (Fig. 6B), but the resulting best-fitting parameters differ dramatically, especially in the control parameters. This suggests that the model cannot adequately capture the differences between the visual-only and visuo-vestibular conditions. This is surprising, given the extensive body of work on dynamic visuo-vestibular integration over the past decades (e.g., Raphan/Cohen, Merfeld, Laurens, and Karmali).

Due to these limitations, I do not believe the modeling component of the manuscript is ready for publication. While the model is grounded in sound hypotheses and the established formalism of optimal control, it is puzzling that it fails to reproduce basic phenomena, such as the vestibular system's contribution to high-frequency self-motion perception. Furthermore, the approach used for fitting “Model 2” raises significant concerns. Addressing these issues will require substantial additional work.

Major Comments

Model Fitting:

As noted, “Model Fit 1” fails to explain the differences between visual-only and visuo-vestibular conditions. In “Model Fit 2,” the control cost parameter r is effectively set to zero in the visuo-vestibular condition (and to 1 in the visual condition). In essence, this approach essentially differences in performance across sensory contexts to changes in the control policy, which is illogical, and fails to capture the underlying sensory differences.

In view of this, the claim in the abstract, “Using computational modeling, we demonstrate that this enhanced performance is partially due to the shorter delay of the vestibular modality,” appears to be unsupported. In fact, the last paragraph on page 16 of the Discussion seems to explicitly contradict this statement.

The authors also use a parameter σs, which scales all sensory noise collectively. This parameter raises several concerns:

(1) It needs to be clearly defined in the text, as the only explicit explanation appears in the equation on page 18 (σs.diag([0.02,0.2,1,0.02,0.2,1])).

(2) It is illogical that all sensory noise, including vision and proprioception, should change by a factor of 9 in the visual-only condition. Moreover, it is odd that noise levels are lower in the visual-only condition, where human performance is worse.

(3) The default value of vestibular noise (σva=1m/s2) appears quite high, which may explain why the model struggles to explain the vestibular contribution to steering.

These points raise concerns about the grid search procedure, where all parameters were scaled by a single factor. It is possible that an insufficiently explored parameter space or an unrealistic parameter combination is driving the observed results.

Based on this, I would recommend that the authors revisit the model design and/or fitting process. While the model shows promise, the current conclusions, based on the fitting approach in Model 2, cannot be accepted.

Minor Comments

Page 1: The statement, “Although studies on multisensory integration indicate that sensory inputs are combined optimally based on their precision, these findings have largely been derived from static experimental settings,” overlooks previous work on Kalman filters and Bayesian estimators for dynamic self-motion perception (visual and vestibular).

Fig. 1D: The labels on the time axis are all 0.

Page 5, second paragraph: I imagine that the sled limits (0.4 m left, 0.6 m right) simulate the driver’s offset position relative to the car's center?

Page 7: The sentence, “Due to this choice, our perturbation is manually mapped to the acceleration state in the true system via Vϵ” is unclear.

Based on equation (5), the force exerted on the wheel (fw) is equivalent to the acceleration of the control signal. While this is clear based on the equations, the authors should explain this explicitly in the text. Otherwise, readers may find the interchangeable use of “steering force” and “acceleration” confusing.

Figure 3: This figure is critical for understanding the model's predictions, but is very difficult to read. Please add legends directly within the panels.

Page 11: The statement, “Combined, this indicates that the primary factor influencing the frequency response characteristics of the model is the controller, while the neuromuscular system and the estimator play a lesser role,” might be influenced by the use of high control costs in all simulations. Consider setting the control cost to zero, and then explore the effects of sensory noise and delay (for the visual and vestibular modality independently).

Fig. 4B: The Y-axis labels are inconsistent, with intervals alternating between 0.02 and 0.03.

Table 1: There is a typo in the description of the parameter λ (vehicle).

Reviewer #2: The authors study multisensory integration in a dynamic environment requiring to take into account time-varying sensory signals and, therefore, their respective delays. Participants were asked to counteract lateral perturbations to keep a virtual car on the road. The perturbations consisted of a pseudo-random sum of sines added to the speed signal. Two conditions were tested: visual only or visuo-inertial, in which the participants' seats were moved laterally, adding vestibular information to the sensory feedback on the perturbations. It is shown, based on gain and phase, that the model with multimodal estimates performs better and reproduces the participants’ behaviour qualitatively.

I found the manuscript very well written and organized. It makes an interesting and important contribution to the field. In my opinion, the paper could be published as is. I have only a number of questions or suggestions to clarify the text and explore alternative hypotheses:

1- I fully agree with the author’s basic premise that standard models of multisensory integration that only consider variance do not generalise to dynamic conditions because dynamics and delays matter. I believe that the authors demonstrate successfully that their model captures some aspects of behaviour, which supports the hypothesis that vision and vestibular signals are combined dynamically, but was there any evidence against a non-dynamic weighting scheme? In Crevecoeur et al. (2016, cited), the constant saccadic reaction time with or without vision was not compatible with linear combination of vision and proprioception. Could the authors explore if or how participant’s behaviour was incompatible with static multisensory integration?

2- I found that Figure 3 was difficult to interpret: what is the most important piece of information? Is it that the orange traces in the bottom right figure are different from the blue ones, showing the effect of vestibular input? If I interpret correctly, the benefit of vestibular was to reduce the phase of the response, meaning that the estimate was closer to the state, enabling “less-out-of-phase”. If this is visible on traces in the temporal domain, adding such traces may help the reader navigate through these plots.

3- I had a question regarding the treatment of the perturbation signal (epsilon_t), which was treated as a noise term. Alternatively, it could be considered as constant or even dynamical input signal. In particular, if participants noticed that perturbation followed sinusoidal traces (at least for some frequencies), this knowledge can be added to the model. Can the authors comment on that? What would be the consequence of adding sinusoidal dynamics to epsilon in the model? Can the authors discuss the impact of considering epsilon as noise?

4- Equation 19: I was surprised to see that the prediction noise (xi_t) appears in the term multiplied by the Kalman gain. If the controller/estimator structure knows this noise term, it can be subtracted and removed. Typically, prediction noise does not impact the terms dependent on Kalman gains (see Eqns. 5.1 in Todorov, 2005, ref 33). If there is a genuine argument for the double appearance of xi_t, it should be better justified.

5- Figure 5: the arbitrary offset in the ordinate axis (0.08) produces an inflated visual effect size. Pls consider alternative data representation.

6- Page 8: it is written that the vestibular feedback delay was 10ms, whereas the appendix has an observability matrix H that sits at an index larger than the proprioceptive delay (Eqn. A.11). Could the authors check/clarify?

7- Just before Eqn. 12 it is written that the proprioceptive delay was 0ms, however this seems inconsistent with the physics of the system. In the condition including lateral displacement, the proprioceptive system will also convey information about motion. This seemd aslo inconsistent with Eqn. A.11, which shows delta_prop > 0. Pls clarify.

8- Eqn. A.11: Pls consider adding indices to the “zeros” to show that these are block matrices, and the product with the time series of state vector is block-wise.

Reviewer #3: The study addresses an important question, which is multisensory integration for active, dynamic behaviors. The experimental methods, modeling and analysis are well-developed and clearly presented with appropriate care taken to identify assumptions and caveats. I have only minor comments/suggestions.

Suggest line numbers for easier referencing in future

Introduction, line 3: I’m not familiar with this designation of optic flow processing pathways as the ‘optokinetic system’. Optokinetic is most often used in the context of reflexive eye movements and may be confused with specific pathways involved in generating those reflexes.

Figure 1. Based on the depiction here, the visual stimulus does not involve optic flow. Instead, this is more of an object motion stimulus, i.e. how is the car moving relative to the static visual background. This is much less ecological than if a first-person view of the road had been used. Some words to acknowledge this observation (and perhaps a justification of this choice) are warranted given the large literature on visual-vestibular integration most often uses optic flow. This should also be acknowledged in the interpretation offered in the discussion section. The observed importance/effect of visual latency may have also been observed if optic flow has been used, so the result may generalize. Nevertheless, I think the non-ecological nature of the stimulus should be acknowledged.

What is the system latency for the impact of a steering command on the movement of the car and/or platform? Is this latency accounted for in the model? If not, what are possible consequences?

Page 6, bottom: ‘Hence, correcting for the perturbation relies on feedback from the visual and/or mostly the vestibular system’ Why use the word ‘mostly’ here? I realize that results suggest vestibular cues provide an advantage, but that observation does not belong here in the methods.

Page9, last sentence before Model Simulations: the further dyanmics… it would be worth expanding this observation and explaining in detail whey these parameters are the ones that are expected ot impact dynamics…

Page 10/11: Much of the section on model simulations reads like a results section; suggest moving this text and figures to the results.

Figure 4: I think that the bars in figure 4 B are superimposed (i.e. the yellow is continuous behind the blue bar), but these could easily be confused for a stacked bar plot. It is worth clarifying.

**Have the authors made all data and (if applicable) computational code underlying the findings in their manuscript fully available?**

Reviewer #1: Yes

Reviewer #2: Yes

Reviewer #3: **No: **However, the authors have provided assurance that code and data will be shared upon acceptance.

PLOS authors have the option to publish the peer review history of their article (what does this mean?). If published, this will include your full peer review and any attached files.

Reviewer #1: No

Reviewer #2: **Yes: **Frédéric Crevecoeur

Reviewer #3: No

**Figure resubmission:**
---

## [Decision Letter · Decision Letter 1]

24 Jun 2025

Dear Mr. Liu,

We are pleased to inform you that your manuscript 'Visuoinertial and visual feedback in online steering control' has been provisionally accepted for publication in PLOS Computational Biology.

Best regards,

Gunnar Blohm, Ph.D.

Academic Editor

PLOS Computational Biology

Daniele Marinazzo

Section Editor

PLOS Computational Biology

Reviewer's Responses to Questions

**Comments to the Authors:**

Reviewer #1: I thank the authors for addressing my concerns by thoroughly revising the model fitting procedure; and I have no further comment. Please note that Table 1 was not visible in the final version of the article you submitted (but I could see it in the red-lined version).

Reviewer #2: The authors have addressed my previous comments and I have no further recommendation.

Reviewer #3: The authors have satisfactorily addressed my comments.

**Have the authors made all data and (if applicable) computational code underlying the findings in their manuscript fully available?**

Reviewer #1: Yes

Reviewer #2: Yes

Reviewer #3: Yes

PLOS authors have the option to publish the peer review history of their article (what does this mean?). If published, this will include your full peer review and any attached files.

Reviewer #1: No

Reviewer #2: No

Reviewer #3: No

---

## [Editor Report · Acceptance letter]

PCOMPBIOL-D-24-02019R1

Visuoinertial and visual feedback in online steering control

Dear Dr Liu,

I am pleased to inform you that your manuscript has been formally accepted for publication in PLOS Computational Biology. Your manuscript is now with our production department and you will be notified of the publication date in due course.

With kind regards,

Anita Estes
